# The Current State of Bispecific Antibodies and T-Cell Directed Therapy in NHL

**DOI:** 10.3390/cancers17071192

**Published:** 2025-03-31

**Authors:** Austin Kordic, Tycel Jovelle Phillips, Jonathan Weiss

**Affiliations:** 1City of Hope Comprehensive Cancer Center, Department of Hematology and Hematopoietic Cell Transplantation, Division of Lymphoma, Duarte, CA 91010, USA; akordic@coh.org; 2Rogel Comprehensive Cancer Center, University of Michigan-Ann Arbor, Ann Arbor, MI 48109, USA; weissjon@med.umich.edu

**Keywords:** non-Hodgkin’s Lymphoma (NHL), diffuse large B-cell Lymphoma (DLBCL), follicular lymphoma (FL), mantle cell Lymphoma (MCL), relapsed/refractory lymphoma, bispecific antibodies (BsAbs), CAR-T cell therapy, immunotherapy

## Abstract

This is a review article detailing available evidence on the use of CAR-T cell therapy and bispecific antibodies (BsAbs) in non-Hodgkin’s lymphoma—specifically diffuse large B-cell lymphoma, follicular lymphoma, and mantle cell lymphoma. The review hopes to detail where CAR T-cell therapy and BsAbs fit in our current treatment paradigm, as well as foreshadow what is to come in the near future.

## 1. Introduction

Non-Hodgkin lymphoma (NHL)is the most common hematological malignancy in adults, with both indolent (i.e., non-aggressive) and aggressive presentations. Diffuse large B-cell lymphoma (DLBCL), an aggressive lymphoma, accounts for most NHL cases. Although 5-year survival rates in the first-line setting for DLBCL range from 60% to 70%, up to 50% of patients are refractory to or relapse after frontline (1L) treatment [1]. In comparison, indolent lymphomas present a unique issue due to the incurable nature for the majority of those afflicted despite several emerging treatment options. A myriad of novel T-cell directed immune based therapies have been developed for patients with both relapsed/refractory (r/r) aggressive and indolent lymphomas. These novel therapies include bispecific antibodies (BsAbs) and chimeric antigen receptor (CAR) T-cell therapies. This review aims to discuss the past, present, and future of immune based therapies in NHL, as well as how novel immune based therapies fit in our current treatment strategies for DLBCL, follicular lymphoma (FL), and mantle cell lymphoma (MCL).

## 2. DLBCL: Historical 

In 1995, the phase III PARMA study established high-dose chemotherapy (HDC) followed by autologous stem cell transplantation (ASCT) as the standard of care for chemosensitive relapsed NHL in the pre-rituximab era. This study produced an impressive overall response rate (ORR) of 84% in addition to 5-year event-free survival (EFS) of 46% and overall survival (OS) rate of 53% [2]. In 1998, Coiffier et al. published the first phase II study demonstrating the efficacy of rituximab—a chimeric CD20 monoclonal antibody—in relapsed or refractory, intermediate- to high-grade B-cell NHL [3]. These findings led to the development of the LNH98-5 study in 2002, which first evaluated the use of chemo-immunotherapy in DLBCL combining rituximab with cyclophosphamide, doxorubicin, vincristine, and prednisone (CHOP) chemotherapy in elderly patients (60–80 years old) with previously untreated DLBCL. The study found a significant improvement in complete responses (CR), event-free survival (EFS), and overall survival (OS) in patients treated with R-CHOP compared to CHOP alone, notably without a clinically significant increase in toxicity [4,5]. This established a new standard of care using chemo-immunotherapy in the front-line treatment of DLBCL.

Subsequent studies in the rituximab era have failed to redemonstrate the success of the PARMA study in patients with chemosensitive r/r DLBCL with prior rituximab exposure who are treated with HDC/ASCT. In 2010, the phase III CORAL study compared rituximab-based salvage regimens (R-ICE vs. R-DHAP). Those with a favorable response proceeded to HDC/ASCT. The study demonstrated a median overall survival (OS) of 10.0 months with a 3-year PFS of 53% and a median time between ASCT and progression of 7.1 months. The median OS was shorter among patients who relapsed within 6 months compared with those relapsing at least 12 months after ASCT [6]. In 2017, the ORCHARRD study evaluated a similar subset of patients using rituximab-based chemoimmunotherapy followed by ASCT. This study reported 2-year PFS and OS rates of 26% and 38%, respectively [7]. These studies illustrated the need for alternative salvage therapies in patients with prior rituximab exposure, as outcomes are poor, especially for those who have early treatment failure. The need for alternative treatment strategies aside from repetitive chemo-immunotherapy has paved the way for the exploration of other treatment modalities. Historical evidence of long-term responses following allogeneic transplantation in hematological malignancies led to the discovery of novel treatments that harness the natural power of our immune system. These advances led to the introduction of CAR-T-cell therapy and bispecific antibodies, which have revolutionized the management of r/r DLBCL in the post-rituximab era. 

## 3. DLBCL and CAR-T Cell Therapy

The introduction of CD19-directed chimeric antigen receptor CAR-T-cell therapy has been the most impactful advance in the management of patients with early-relapsing DLBCL. CAR-T-cell therapy involves the collection of a patient’s own T-cells, via leukapheresis, which are then genetically modified in a lab to better recognize and respond to the CD-19 expressing lymphoma cells. Lymphodepleting chemotherapy is then administered followed by a single intravenous infusion of the CAR-T cells back into the patient where they expand in number and begin to attack the lymphoma. 

The first approvals for CAR-T products for the management of r/r DLBCL were in the 3^rd^-line setting (Table 1). In 2017, the FDA approved the first CAR-T-cell therapy - axicabtagene ciloleucel (axi-cel; Yescarta) - for the treatment of r/r NHL in the third-line setting. This approval was based on promising response rates in the phase II ZUMA-1 trial, which included 101 adult patients with r/r NHL who had progressed on two prior lines of systemic therapy. All patients had chemorefractory disease, with a median of three prior lines of therapy. Most of the patients had primary refractory disease (79% without prior ASCT, 21% relapsed within 12 months of ASCT). A lymphodepleting conditioning regimen of fludarabine and cyclophosphamide was given prior to axi-cel, which was administered as a single infusion of modified autologous T-cells at a target dose of 2 × 10^6^ CAR-positive T-cells/kg. The study showed a promising ORR of 83% with a CR of 58%. The median duration of response was 8.2 months, with a six-month OS of 80%. Both the DLBCL and PMBCL/TFL cohorts showed similar ORR, although CR rates were higher in the PMBCL/TFL cohort (71% vs. 49%) [8]. A five-year follow-up analysis published in May 2023 demonstrated durability of response with ongoing response in 31% of patients, with a median overall survival (OS) of 25.8 months and estimated 5-year OS rate of 42.6% [9]. There were no new serious adverse events or deaths related to axi-cel observed after additional follow-up. Two additional CAR-T products—lisocabtagene maraleucel (liso-cel) and tisagenlecleucel (tisa-cel)—were also approved in the third-line setting for DLBCL. Liso-cel was evaluated in a multicenter study known as TRANSCEND. Patients were eligible if they had received two or more prior lines of therapy. The ORR was 73% and the CRR was 53% [10]. Tisa-cel was evaluated in a large phase II study in patients with relapsed/refractory DLBCL. This study reported an OR rate of 52% and a CR rate of 40% [11]. The promising results from these landmark studies opened the door for CAR-T as an emerging tool in treating chemorefractory r/r NHL in the 2^nd^-line (2L) setting (Table 2).

In 2022, ZUMA-7 was the first large phase III study to compare the safety and efficacy of 2L axi-cel with standard-of-care chemoimmunotherapy followed by HDT-ASCT. This study included 359 adult patients with r/r DLBCL, most of which had primary refractory disease (74% vs. 26% relapsed within 1 year of first line therapy). The study included a mixture of subtypes (16% HGBL [double/triple-hit], 33% double-expressor, 6% MYC rearrangement) and patient age (30% were 65 years of age or older). The primary endpoint of modified event-free survival (mEFS) showed an impressive advantage with axi-cel (8.3 months) vs. SOC (2.0 months) for a hazard ratio (HR) of 0.4 (*p* < 0.0001). The estimated 18-month EFS rates (41.5% vs. 17%) and response rates (83% with 65% CR vs. 50% with 32% CR) also favored axi-cel over SOC (OR, 5.31; *p* < 0.0001). The EFS benefit was consistent across key patient subsets including age ≥ 65 years (HR 0.276), primary refractory versus relapsed disease (HR 0.42 vs. 0.34), and molecular subtypes (HGBL–double/triple-hit; HR, 0.28 vs. double-expressor lymphoma; HR, 0.42). At the time of the interim analysis, the median OS was not yet reached with axi-cel compared with 35.1 months with SOC (HR, 0.73; *p* = 0.027). Safety monitoring was critical due to the risk of cytokine release syndrome (CRS) and neurological toxicity with CAR-T therapy. Grade 3 or higher CRS occurred in 6% of patients with a median time to onset of 3 days and median duration of event was 7 days. In the safety population, tocilizumab and glucocorticoids were administered to 65% and 24% of the patients, respectively. Neurological toxicity with grade 3 or higher immune effector cell-associated neurological toxicity syndrome (ICANS) was noted in 21% of those who received axi-cel with a median time to onset of 7 days and the median duration of event was 9 days. Glucocorticoids were used in 32% of the patients for the management of neurologic events [17].

In 2022, the phase III TRANSFORM trial compared liso-cel to platinum-based immunochemotherapy followed by HDC/ASCT in the 2L setting for early-relapsed or refractory LBCL. The study reported a significant improvement in CR rate with liso-cel compared to SOC (74% vs. 43%l; *p* < 0.0001). In the primary analysis with a 17.5-month median follow-up, median PFS was NR for liso-cel vs. 6.2 months for SOC (HR, 0.4; *p* < 0.0001). Median OS was NR for liso-cel vs. 29.9 months for SOC (HR, 0.724; *p* = 0.0987). When adjusted for crossover, 18-month OS rates were 73% for liso-cel and 54% for SOC (HR, 0.415). Grade 3 cytokine release syndrome and neurological events occurred in 1% and 4%, respectively [19]. The results of these studies supported the use of axi-cel and liso-cel as preferred second-line treatments in patients with primary refractory or early relapsed DLBCL who were determined to be adequate candidates for CAR-T therapy. 

In 2022, the phase III BELINDA study, tisa-cel, to salvage chemotherapy followed by ASCT (SOC) in patients with r/r LBCL within 12 months of first-line therapy. The study found that while response rates were slightly higher in the tisa-cel group (46.3% vs. 42.5%), the study failed to show a benefit, with many patients demonstrating disease progression at week 6 compared to the SOC group (25.9% vs. 13.8). In the tisa-cel group, 5.2% had grade 3 or higher CRS with a median time to onset of 4 days with a median duration of 5 days. Grade 3 or higher neurological events were noted in 1.9% of patients, with a median time to onset of 5 days and median duration of 9 days. While unable to fully verify, the belief is that the failure of this study to demonstrate a benefit within the experimental arm was due to the long median time from leukapheresis to infusion of 52 days, which highlights the challenges of delivering CAR-T therapy in patients with an aggressive disease biology [20].

Older patients present an interesting dilemma due to co-morbidities that might impact tolerance to treatment. To further evaluate how CAR-T-cell might fare in the elderly patient population, liso-cel was evaluated in a phase II PILOT study of 2L patients deemed ineligible for ASCT. The median age of those enrolled in this study was 74 years old. The primary end point was reached with an ORR of 80% and a CRR 54% after a median follow-up of 12.3 months. The study also found that the responses were durable, with a median duration of response (DOR) of 21.7 months with a median PFS of 9 months. Grade 3 CRS and neurological events occurred in 1.6% and 4.9% of liso-cel recipients, respectively, with no grade 4 or higher events [21].

## 4. DLBCL and BsAbs

Another major advancement in management of r/r NHL has been the development of T-cell engaging bispecific antibodies. The CD 20/CD 3 T-cell engager BsAbs are treatment options that have further revolutionized the management of r/r DLBCL. These off-the-shelf drugs have proven essential as they are more readily available without the prolonged production time of CAR-T products. These drugs are administered in what is described as a step-up dosing (SUD) method to reduce toxicity. This method entails starting with a low dose with a pre-defined escalation to the full target dose. Thereafter, patients are treated at pre-specified intervals based on the agent. These drugs have been approved in the 3L setting for patients with r/r DLBCL, including patients who relapsed following CAR-T (Table 1). Similar to CAR-T products, they are now also being explored in earlier lines of therapy [22].

In May 2023, epcoritamab was granted accelerated approval for the treatment of r/r DLBCL in the third-line setting based on the results of the phase II EPCORE NHL-1 study. Epcoritamab, a novel subcutaneous bispecific CD20-directed CD3 T-cell engager, first demonstrated impressive activity in r/r mature B cell lymphomas in a phase I/II dose-expansion study by Hutchings et al. The ORR was found to be 68% with a CR rate of 45% at 12–60 mg doses with a tolerable safety profile [23]. In the expansion portion of the phase ½ study, EPCORE NHL-1, epcoritamab was found to have deep, durable responses with a manageable safety profile when administered as a single agent with ORR 63.1%, and CRR of 38.9% and a median duration of response of 12 months [24]. Epcoritamab was administered subcutaneously in 28-day cycles until disease progression or unacceptable toxicity following weekly SUD during cycle 1 according to the following schedule: 0.16 mg (D1) → 0.8 mg (D8) → 48 mg (D15, D22). This was followed by fixed dosing of 48 mg in subsequent cycles that was administered weekly in cycles 2–3, every 2 weeks in cycles 4–9, and then every 4 weeks in cycle 10 and beyond. In the dose expansion phase of the trial, cytokine release syndrome (CRS) events occurred in 49.7% of patients and were mostly low-grade and had predictable timing, with most events following the first full dose at cycle 1 day 15 (C1D15). To mitigate CRS in dose expansion, epcoritamab was given with steroid premedication and intravenous fluid hydration in addition to SUD as described. Per the prescribed label, patients should be hospitalized for 24 hours after the cycle 1 day 15 dose of 48 mg to monitor for CRS and ICANS.

Glofitamab, a novel intravenous CD20 × CD3 bispecific antibody with bivalency for CD20 on B cells and monovalency for CD3 on T-cells, was granted accelerated approval for the treatment of adult patients with r/r DLBCL after two or more lines of systemic therapy based on the results of a phase I/II study [15]. Of note, the study population included approximately 30% of patients who had received prior CAR-T therapy. The results of the study showed an ORR in 52% with a CR in 39% of patients. The median duration of response of 18.4 months with 68.5% of responders shows a continued response at 9 months. To mitigate the risk of CRS, pretreatment with a single dose of Obinutuzumab 1000 mg IV (a humanized anti-CD20 monoclonal antibody) was given 1 week prior to the start of SUD for glofitamab (C1D1) in addition to corticosteroid (dexamethasone 20 mg) pre-treatment on the day of infusion. Glofitamab is administered once every 3 weeks for a maximum of 12 cycles (approximately 8.5 months), which is distinct compared to the indefinite dosing with epcoritamab. The first dose of glofitamab on the SUD schedule is 2.5 mg given on Day 8 (C1D8), followed by 10 mg on Day 15 (C1D15), followed by the first full dose of 30 mg on Day 21 (Cycle 2 Day 1). Among 145 patients who received glofitamab in the study, the most common adverse events were cytokine-release syndrome (70%), musculoskeletal pain (21%), fatigue (20%), and rash (20%). Cytokine-release syndrome was generally low-grade (52% of patients grade 1, and 14% grade 2).

Epcoritamab and glofitamab have continued to revolutionize the management of r/r DLBCL in the third line setting, including patients who have relapsed following 2L CAR-T. There are ongoing studies evaluating these therapeutics earlier on in a patient’s treatment course. This is particularly important for patients who are not candidates for CAR-T-cell therapy, either due to financial/logistical limitations or patients with an aggressive disease biology who cannot wait multiple weeks for the apheresis and manufacturing process. Given the BsAbs are “off the shelf”, and do not require prolonged manufacturing times, they may be enticing over CAR-T-cell therapy in specific situations. Additionally, some patients live in areas that do not have access to tertiary care centers, and therefore are unable to receive CAR-T-cell. In this situation, BsAbs may be reasonable options given the results published in the STARGLO study, which evaluated patients in the 2L+ setting, coupled with the single-agent activity reported in CAR-T-naïve patients.

BsAbs do have barriers as well. Given the side effect profile (i.e., CRS, neurological toxicity), administering centers must have detailed plans, and strategies to deal with some of these complications. Administering centers must be able to closely monitor patients and administer tocilizumab. Not all community practice centers currently have the capabilities to manage BsAb administration. However, it is much more feasible to implement when compared to CAR T-cell therapy, and given the development of BsAbs in solid tumors, this will likely become a more widely used modality throughout oncology practices. Thankfully, detailed strategies and plans for BsAbs implementation have been published, and can help spread access to these life-saving therapies [25].

The ideal sequencing of treatments for R/R DLBCL would favor using CAR-T prior to bispecific antibody therapies due to the established curative potential of CAR-T. However, there are a combination of factors that can complicate this matter. There is a significant time period required for CAR-T approval, apheresis, production time, the infusion which altogether can last several weeks to a few months. Additionally, some of the products run the risk of being outside of pre-determined specifications, which can add additional time to treatment. Given that a subset of patients can have rapid progression of their disease and or social barriers that prevent them from receiving CAR-T-cell therapy, BsAbs can provide a readily available option for disease control.

## 5. Follicular Lymphoma

Follicular lymphoma (FL) presents a unique clinical challenge due to the indolent but incurable nature of the disease. This key difference compared to DLBCL results in a greater emphasis on reducing treatment-related toxicities, which limited the early implementation of CAR-T in the management of FL. Nonetheless, this treatment modality ultimately demonstrated impressive efficacy and safety in patients with r/r FL in the 3^rd^-line setting (Table 3). 

In 2022, the phase II ELARA study evaluated the efficacy of tisa-cel in adults with r/r FL after two or more treatment lines or who relapsed after autologous stem cell transplant. The primary endpoint was CR rate (CRR) with secondary endpoints of ORR, PFS, OS, duration of response, pharmacokinetics and safety. The primary endpoint was met with a CRR of 69.1% and ORR 86.2%. Within 8 weeks of infusion, rates of cytokine release syndrome were 48.5% with no CRS events of grade 3 or higher. The rate of neurological events was 37.1% with 3% grade 3 or higher. The rate of immune effector cell-associated neurotoxicity syndrome (ICANS) was 4.1% with 1% grade 3 or higher . Notably, there were no treatment-related deaths. The results of this study demonstrated that tisa-cel was safe and effective in extensively pretreated r/r FL, including in high-risk patients [31]. In 2022, the phase II ZUMA-5 evaluated axi-cel in r/r FL and also demonstrated high response rates with durable responses. At the 3-year follow up analysis, the ORR was 94%, CRR 74%, and median PFS was 40.2 months. Notably, there were no reported events of any-grade CRS or neurological toxicity amongst the cohort of patients with r/r FL [32]. In 2023, the phase II TRANSCEND FL study evaluated the safety and efficacy of liso-cel in patients with high-risk (i.e., progression of disease/POD within 24 months) or refractory FL. Liso-cel demonstrated an impressive ORR and CRR of 95.7% (i.e., all patients that responded had a CR). CRS events occurred in 58% of patients with grade 3 or higher CRS events in 1%. Neurological events occurred in 15% of patients with grade 3 or higher events in 2% of patients [33]. 

The relative benefit of CAR-T to BsAbs is distinct in FL compared to DLBCL, given that neither therapy has been shown to be curative in this setting. In light of this, clinicians must consider the convenience, cost, and relative safety of BsAbs compared to CAR-T. In 2022, mosunetuzumab-axgb (Lunsumio) became the first approved bispecific antibody therapy for r/r FL in the third-line setting based on the results of the GO29781 study. This was an open-label, multicenter, multi-cohort study that included 90 patients who had previously been treated with an anti-CD20 monoclonal antibody and an alkylating agent. The main efficacy outcome measure was objective response rate (ORR), which was 80% and 60% CR. With a median follow-up of 14.9 months among responders, the estimated median duration of response (DOR) was 22.8 months, and the estimated DOR rate at 12 months and 18 months was 62% and 57%, respectively. The therapy was administered on a 21-day cycle according to the following schedule: 1 mg on Cycle 1 Day 1, 2 mg on Cycle 1 Day 8, 60 mg on Cycle 1 Day 15, 60 mg on Cycle 2 Day 1, and 30 mg on Day 1 in subsequent cycles. Notably, the study design allowed for the discontinuation of therapy after eight cycles in patients with a complete response. Patients with a partial response (PR) or stable disease (SD) should continue treatment for up to 17 cycles unless they experience progressive disease or unacceptable toxicity. CRS events occurred in 44% of patients (Grade 3–4: 2.5%), primarily during cycle 1 SUD. Neurologic toxicity occurred in 39% (ICANS: 1%), serious infections in 17%, and tumor flare in 4%. The most common Grade 3 to 4 laboratory abnormalities (≥10%) were cytopenias. Notably, no Grade 5 events were reported. This study demonstrated the safety and efficacy of fixed duration mosunetuzumab in patients with relapsed/refractory follicular lymphoma in the 3rd line setting [26]. 

No clinical trials have directly compared BsAbs to CAR-T-cell therapy in FL, therefore, we have no concrete evidence to tell us what therapy is “better” or should be given first. As noted above, the lack of curative intent treatment in this setting places a stronger emphasis on patient safety when comparing our treatment options. Accessibility remains a major concern given that CAR-T-cell therapy can only be given at a specialized/tertiary referral centers with well-established protocols for cellular therapies. These centers are often located in major metropolitan centers and university hospitals, which the potential impact of CAR-T cell therapy in rural or underserved populations. The biggest challenge for administering CAR-T-cell products in the community setting is developing a program that has the appropriate clinical expertise and infrastructure to manage both drug delivery and management of serious adverse effects. It requires inpatient and outpatient arrangements in addition to expert subspecialty support in the hospital for patients who have treatment-related complications [34]. It would be reasonable to utilize a CAR-T-cell product in patients with FL with concern for transformation to DLBCL, given that it has curative potential in this setting as noted above. Additionally, patients with progression of disease within 24 months of frontline therapy (i.e., POD24) may benefit from CAR-T-cell therapy due to the higher rate of histological transformation in these patients. Outside of those with transformed DLBCL, the true impact of POD24 is unclear. As such, BsAbs likely may remain reasonable options in these patients give a more favorable toxicity profile in general, with lower rates of neurotoxicity, CRS, and inpatient complications. In FL, the decision to pursue CAR-T-cell therapy or BsAB is generally made on a case-by-case basis with all of these factors in mind.

## 6. MCL

Mantle cell lymphoma (MCL) is a rare lymphoma that has historically demonstrated much shorter remission and overall survival compared to other indolent lymphomas, particularly in the relapsed/refractory setting [35]. The frontline treatment for MCL has historically been chemo-immunotherapy with or without HDC/ASCT and the more recent incorporation of covalent Bruton’s tyrosine kinase (BTK) inhibitor therapy. Bispecific antibody and CAR-T cell therapies have now been approved for the management of r/r MCL (Table 4). 

In 2020, the results of the phase II ZUMA-2 study led to the first major breakthrough of cellular therapies in the management of r/r MCL. This study led to the approval of brexucabtagene autoleucel (brexu-cel), an autologous CD19-directed CAR-T-cell therapy, in patients with r/r MCL. Patients were required to have received prior anthracycline- or bendamustine-containing chemotherapy, an anti-CD20 monoclonal antibody, and BTK inhibitor therapy. Patients underwent leukapheresis and optional bridging therapy, followed by conditioning chemotherapy and a single infusion of KTE-X19 at a dose of 2 × 10^6^ CAR-T-cells per kilogram of body weight. The primary efficacy analysis showed that 93% had an objective response with 67% having a complete response. At a median follow-up of 12.3 months, 57% were in remission. At 12 months, the estimated PFS and OS were 61% and 83%, respectively. Common adverse events of grade 3 or higher were cytopenias (94%) and infections (32%) with two grade 5 infectious events. Grade 3 or higher CRS and neurologic events occurred in 15% and 31% of patients, respectively, with no grade 5 events [38]. Additionally, Wang et al. published a multicenter study in JCO that reported on clinical outcomes in the “real world setting” in which 189 patients underwent leukapheresis, 79% of which would not have met ZUMA-2 eligibility. Best overall and complete response rates were 90% and 82%, respectively, with a 12-month PFS of 59%. Grade 3 or higher CRS and neurotoxicity occurred in 8% and 32%, respectively. In univariable analysis, high-risk simplified MCL international prognostic index, high Ki-67, *TP53* aberration, complex karyotype, blastoid variant, and bendamustine exposure within 24 months of leukapheresis had shorter PFS [36].

Lisocabtagene maraleucel (liso-cel), another autologous CD19-directed CAR-T-cell therapy, has also been evaluated in r/r MCL after two lines of prior therapy. In 2020, the Transcend NHL 001 study evaluated liso-cel in patients with r/r MCL, with a reported ORR of 86.5% and a CRR of 74.3%. In patients with high-risk features (i.e., blastoid morphology, high Ki-67, and TP53 mutated), the ORR were greater than 70% in each subgroup. The responses were quite durable with a median duration of response of 15.7 months after a median follow up of 22.8 months. The median OS in those patients with a CR was 36.3 months. CRS events were limited (42% any grade, 2% grade 3 or higher), while higher-grade neurological toxicity was relatively more common (30% any grade, 10% grade 3 or higher), albeit less than what was observed in the ZUMA-2 study. There were no Grade-5 CRS or ICANS events reported. One patient died from diffuse alveolar hemorrhage following a dose of 50 × 10^6^ CAR-T cells [39]. The results of this study established liso-cel as an excellent treatment option in r/r MCL, especially in light of the favorable safety profile of liso-cel compared to bruxu-cel in this patient population. It should be noted that the results in patients with prior treatment with BTK inhibitors were less impressive. In this patient population, the study reported a median DOR, PFS and OS of 5.3 months, 6.1 months and 11.1 months, respectively. Further maturation of the data is needed to obtain further clarity on the role of this treatment modality with the incorporation of BTKi therapy in the frontline setting of r/r MCL. 

Similar to FL and DLBCL, BsAbs have also established a role in the management of r/r MCL (Table 4). Glofitamab was evaluated in 3L setting for r/r MCL in the NP30179 study, which was a phase I/II clinical trial that included patients who had received two prior lines of therapy (including a BTK inhibitor). In this pivotal study, glofitamab was found to have an ORR and CRR of 85.0% and 78.3%, respectively, amongst all patients versus 74.2% and 71.0%, respectively, in patients with prior BTKi therapy. The median DOR was 16.2 months. Two dosing cohorts of obinutuzumab pretreatment were studied (1000 mg vs 2000 mg), both of which were administered 7 days before the first glofitamab dose (C1D1). The glofitamab SUD schedule was as follows: 2.5 mg (C1D8), 10 mg (C1D15), followed by a target dose of 16 mg or 30 mg every 21 days starting with Cycle 2 Day 1 (C2D1) for up to 12 cycles. As expected, CRS was the most common adverse effect, with any-grade CRS events occurring in 70.0% of patients with a lower incidence reported in the 2000 mg cohort of obinutuzumab pretreatment compared to the 1000 mg cohort. Four adverse events led to drug discontinuation, all of which were infections [37]. In 2024, Budde et al. reported extended follow up of phase I/II study data using mosunetuzumab monotherapy in patients with r/r NHL, including a small cohort of patients with r/r MCL. While the number of patients was limited, the ORR and CRR are 3/13 (30.8%) and 3/13 (23.1%), respectively [40]. While single agent efficacy data in patient with r/r MCL were not impressive, further studies have demonstrated more promising response rates when combined with the antibody-drug conjugate (ADC) polatuzumab vedotin resulting in an ORR of 75% and CRR of 70% [41]. These results paved the way for additional combinations of BsAbs and ADCs (i.e., mosunetuzumab + polatuzumab-vedotin, etc.), which will further improve our treatment options for r/r MCL. 

Unfortunately, while the CAR-T and BsAbs that have been discussed have shown great promise in the r/r setting, there are still a portion of patients whose disease is “resistant” and fails to respond or whose response is short-lived. Mechanisms for resistance are currently under investigation, but may be due to an unfavorable tumor microenvironment (TME), specific genetic alterations (e.g., TP53 mutation), or loss of cell surface antigens (e.g., CD20 and CD19). Perhaps future methods to enhance, or alter the TME (i.e., effector T-cell function repolarization), may help improve outcomes. Additionally, targeting an alternative cell surface antigen (e.g., CD22, CD30, BAFF-R etc.) or using multiple target antigens (tri-specific antibodies) may result in improved outcomes in these challenging clinical scenarios.

## 7. Future Directions

The future of CAR-T-cell therapy and BsAbs is bright in NHL. Over the coming months to years, we will see new combinations of therapies, new cellular targets, and novel immune-based strategies all together. For example, CD22-directed CAR-T-cell therapies are being studied in patients with DLBCL who have relapsed following CD19-directed CAR-T-cell therapy. In one phase I study including this high-risk patient population, CD22-directed CAR-T cell therapy resulted in an ORR was 68% with a CRR of 53%. The novel product had no patients develop a dose limiting toxicity, including no reported grade 3 or higher CRS [42]. There is an ongoing phase II study to further evaluate this CD22 CAR-T-cell product in r/r DLBCL [43]. Additionally, clinical studies are ongoing using CAR-T products targeting B-cell activating factor-receptor (BAFF-R) in hematologic malignancies [44]. Odronextamab is yet another CD20 × CD3 BsAb that has shown promise in r/r DLBCL and FL. In the Phase II ELM-2 study, patients with r/r DLBCL were treated with IV odronextamab in 21-day cycles with a reported ORR and CRR of 52% and 31%, respectively [45]. 

“Trispecific Antibodies” or TsAbs are the newest frontier of emerging treatment modalities in the management of r/r NHL. JNJ-79635322 is a TsAbs developed for management of multiple myeloma, with a novel BCMA, GPRC5D, and CD3 T-cell redirecting antibody [46]. T-cell receptor (TCR)-engineered T-cell therapy is an active area of investigation as well—which may possess more durable signal activation than standard CAR-T-cells. For example, ET019003 cells are novel anti-CD19 gamma/delta TCR T-cells that were evaluated in r/r DLBCL with a reported ORR of 87.5%, CRR of 75%, and 3-year OS of 75%. One patient with primary CNS lymphoma had a CR ongoing at 3 years [47]. CAR natural killer (NK) cell therapy is another area of active investigation. FT596 is an “off-the-shelf” CD19-directed CAR NK cell product that was evaluated in a multi-center phase I trial in patients with r/r B-cell lymphomas. This study showed that FT596 was well tolerated and had overall favorable responses, including patients with prior CD19-directed CAR-T-cell therapy [48]. These are just a few examples of the exciting targets and therapies that are making their way through the research pipeline and into clinical practice.

Further research in this disease space will hopefully clarify the optimal sequencing of T-cell mediated therapies. For example, high-risk patients with DLBCL (i.e., high International Prognostic Index (IPI) score of 3–5, double/triple hit mutation (DHL/THL), or PET scan-positive after two cycles of chemotherapy) have lower PFS and OS rates using standard rituximab-containing chemoimmunotherapy compared to low-risk patients. In November 2022, the phase II ZUMA-12 study compared axi-cel with standard of care chemoimmunotherapy in the first line setting in patients with high-risk DLBCL. This study showed a significant benefit of axi-cel compared to chemoimmunotherapy in patients with DHL/THL (CR: 78% vs. 40%), PET-2 positive (CR 78% vs. 64%), and IPI 3–5 (CR 81% vs. 74%) [49]. Additionally, multiple ongoing studies are exploring bispecific antibodies in earlier lines of therapy in several subtypes of lymphoma. Epcoritamab has shown promise when used in combination with R-CHOP chemotherapy in the frontline setting for patients with newly diagnosed DLBCL. The results from an ongoing phase 1/2 study in high-risk pts with newly diagnosed DLBCL (EPCORE NHL-2 arm 1; NCT04663347) show that epcoritamab + R-CHOP has promising efficacy and a manageable safety profile in high-risk patients with IPI 3–5 disease. Among efficacy-evaluable pts (n = 31), the ORR was 100% and complete metabolic response (CMR) rate was 77%. CRS events occurred in 52% of patients and were mostly low-grade, had predictable timing, and did not require discontinuation of treatment. These data were encouraging for the prospect of incorporating bispecific antibody therapy into the frontline setting of treatment for DLBCL [50]. SKYGLO is a phase III study evaluating the incorporation of BsAbs with chemo-immunotherapy in the frontline management of DLBCL. This study has combined glofitamab plus polatuzumab vedotin-rituximab-cyclophosphamide-doxorubicin-prednisone (Glofit-Pola-R-CHP) in front-line DLBCL, the results of which will be eagerly awaited [51].

BsAbs are also being studied in the frontline setting of FL and MCL. Mosunetuzumab was evaluated in a phase II, multi-center trial in patients with untreated FL. This study reported a best overall response of 96% and a CRR of 81% [52]. Similarly, the combination of epcoritamab plus bendamustine/rituximab (BR) chemo-immunotherapy was evaluated in the frontline management of FL. This study reported a 96% CRR with durable responses and evidenced by 87% remaining in a CR at 30 months [53]. Additionally, chemotherapy-free combinations of lenalidomide + mosunetuzumab, and lenalidomide + epcoritamab have shown promising results in this space [54,55]. The ‘GLOBRYTE’ trial is an ongoing phase III study comparing glofitamab to physicians choice chemo-immunotherapy in r/r MCL, which will likely provide additional rationale for the use of BsAbs in this high risk patient population [56]. Further studies will seek to incorporate these agents into the frontline treatment of MCL and may one day offer the possibility of curative intent therapy. 

Some of the main questions going forward are as follows: (1) How should we sequence these therapies? (2) How should we combine these therapies? (3) What other cellular targets should we pursue? Overall, it is an exciting time in the field of NHL with the benefits of these treatments continuing to push the envelope for remission and survival for patients NHL.

## 8. Conclusions

Over two decades ago, we witnessed the way in which CD20 monoclonal antibodies revolutionized the management of B-cell NHL. Over the last decade, T-cell-directed therapies (BsAbs and CAR-T-cell) have been having an equally exciting and profound impact on the field. The “off-the-shelf” availability of BsAbs combined with more robust clinical infrastructure and clinician comfort using both BsAbs and CAR-T cell therapies should result in even broader access to these powerful new treatment modalities in the coming years. Further refinement of novel targets, combinations, and sequencing of these therapies will continue to unleash the limitless potential that is just only being tapped. 

## Figures and Tables

**Table 1 cancers-17-01192-t001:** CAR-T-cell and BsAbs therapies—third line and beyond—DLBCL.

	ORR/CRR	Median DOR	Grade 3 (or Higher) CRS	Grade 3 (or Higher) Neuro Event	Other
**Axicabtagene Ciloleucel** [9,12]	83%/58%	11.1 months	11%	32%	Median OS: 25.8 months 5-year OS rate: 42.6%
**Lisocabtagene Maraleucel** [10,13]	73%/53%	23.1 months	2%	10%	Median OS: 27.3 months
**Tisagenlecleucel** [11,14]	52%/40%	Not reached	22%	12%	OS at 18 months: 43%
**Glofitamab** [15]	52%/39%	18.4 months	4%	3%	Estimated 12-month OS 50%
**Epcoritamab** [16]	63.1%/40.1%	17.3 months	3.2%	0.6%	Median OS 18.5 months

**Table 2 cancers-17-01192-t002:** CAR-T-cell therapies—second line—DLBCL.

	OS	PFS/EFS	Bridging Therapy	Safety (Grade 3 or Higher)
**Axicabtagene Ciloleucel** [17,18]	-Estimated 4 yr OS 54.6% -OS benefit over SOC	-Estimated 4 yr PFS 41.8% -Median PFS 14.7 months -PFS benefit over SOC	-Glucorticoids allowed	CRS: 6% Neuro event: 21%
**Lisocabtagene Maraleucel** [19]	-12-month OS 79.1% -OS benefit over SOC	-Median EFS 10.1 months -EFS benefit over SOC	-Glucocorticoids allowed	CRS: 1% Neuro event: 4%
**Tisagenlecleucel** [14,20]	-No difference in OS between tisagenlecleucel and SOC	-No difference in EFS between tisagenlecleucel and SOC	-Chemotherapy allowed	CRS: 5.2% Neuro event: 1.9%

CRS-Cytokine Release Syndrome; DLBCL-Diffuse Large B-cell Lymphoma; EFS-Event Free Survival; OS-Overall Survival; PFS-Progression Free Survival; SOC-Standard of Care.

**Table 3 cancers-17-01192-t003:** CAR T-cell and BsAbs Therapies–r/r FL.

	ORR/CRR	Median DOR	Grade 3 (or Higher) CRS	Grade 3 (or Higher) Neuro Event	Other
**Mosunetuzumab** [26]	80%/60%	22.8 months	2%	0%	Median PFS 17.9 months
**Tisagenlecleucel** [27]	86.2%/68.1%	Not reached	<1%	<1%	24-month PFSwas 57.4%
**Axicabtagene Ciloleucel** [28]	94%/79%	Not Reached	7%	19%	18-month OS 87.4%
**Lisocabtagene Maraleucel** [29]	97%/94%	Not Reached	1%	2%	12-month PFS rate 91%
**Epcoritamab** [30]	82%/62.5%	---	2%	0%	Estimated PFS at 18 months 49%

**Table 4 cancers-17-01192-t004:** CAR-T-cell and BsAbs therapies—r/r MCL.

	ORR/CRR	Grade 3 (or Higher) CRS	Grade 3 (or Higher) Neuro Event	Other
**Lisocabtagene Maraleucel [10]**	87%/74.3%	2%	27%	Estimated DOR at 1 year was 55%
**Brexucabtagene autoleucel [36]**	93%/67%	15%	31%	Median follow-up of 12.3 months, 57% were in remission
**Glofitamab [37]**	85%/78%	14%	(No ICANS)	Median DOR 16.2 months

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
