# Peer review of "The Current State of Bispecific Antibodies and T-Cell Directed Therapy in NHL"

_cancers, 2025, doi:10.3390/cancers17071192_

Round 1
Reviewer 1 Report
Comments and Suggestions for Authors
Dear Authors,
Congratulations on your excellent manuscript in this area.
I would like to add several comments,
CART therapy of DLBCL in page 4, I doubt BsAb can be used for disease control/Bridging therapy before CART.
Follicular lymphoma in page 5,
I personally believe BsAB takes precedence over CART therapy in terms of sequence of therapy although lack of comparative data except high risk case - POD24 where I prefer CART therapy.
Mantle cell lymphoma in page 6,
Correction - Brexucel was approved for 2nd line and further.
Author Response
I would like to add several comments,
CART therapy of DLBCL in page 4, I doubt BsAb can be used for disease control/Bridging therapy before CART. (While we did not explicitly say this in the text but Crochet et al Blood (2024) 144 (3): 334–338 does support that this modality does not impair efficacy and as such can be evaluated as a potential bridge to CAR-T)
Follicular lymphoma in page 5,
I personally believe BsAB takes precedence over CART therapy in terms of sequence of therapy although lack of comparative data except high-risk case - POD24 where I prefer CART therapy. (A sentence about POD24 was added but would say for those who have confirmation that there is a lack of tDLBCL, an open discussion w/ patient on the risks/benefits is warranted given the uncertainty on outcomes for POD24 based on data from
Muntañola A, Mozas P, Mercadal S, et al. Early progression in follicular lymphoma in the absence of histological transformation or high-risk follicular lymphoma international prognostic index still has a favourable outcome. Br J Haematol. 2023;200(3):306-314.)
Mantle cell lymphoma in page 6,
Correction - Brexucel was approved for 2nd line and further. (I changed to r/r MCL)
This is a well-written review manuscript, focusing on summarizing the past, present, and future of immune based therapies in NHL, including DLBCL, FL and MCL.
Major point
- The CAR-T therapies and bispecific antibodies under clinical investigation was largely missing. This information is critical for the readers to understand the potential direction, especially those with new strategies or new targets. (We have added to the section on future directions, I added a few thing [NK cell CARs, TCR-T therapies])
- The preclinical development using new strategies and targets is also missing.(Thanks that is an interesting component of the future of these agents but is outside of the scope for this article.
- TCR-T therapies not mentioned. (We have added information to this section but overall given the limited data noted thus far we have limited the information on these treatments.)
minor point:
- first line: Reiew --> Review (Fixed)
- Page 4: due --> do (Fixed)
This paper provides a comprehensive review of novel T-cell-directed therapies for patients with relapsed/refractory aggressive and indolent Non-Hodgkin Lymphomas (NHL). The authors offer a detailed discussion of various clinical trials involving CAR T-cells and bispecific antibodies.
This is a timely and relevant review that highlights both the benefits and challenges of these emerging therapies. Overall, the paper is well-written, but some formatting issues should be addressed.
Currently, the paper includes only two tables summarizing CAR T-cell therapy outcomes for DLBCL, with no corresponding data for other lymphoma types. It would be beneficial to include additional tables summarizing results for other NHLs. Furthermore, tables should be integrated within the main text rather than placed at the end of the manuscript.
(We have added two additional tables and embedded them with appropriate section)
Additionally, a comparative table outlining the key differences between CAR T-cell therapies and bispecific antibodies would enhance the paper's clarity and usefulness. (I included both BsAbs and CAR in the tables)
To improve readability, the text should be better structured by dividing long paragraphs into clearly defined sections. (I broke it up into a few more sections. Specifically the DLBCL section—as this was too long)

Reviewer 2 Report
Comments and Suggestions for Authors
This is a well-written review manuscript, focusing on summarizing the past, present, and future of immune based therapies in NHL, including DLBCL, FL and MCL.
Major point
- The CAR-T therapies and bispecific antibodies under clinical investigation was largely missing. This information is critical for the readers to understand the potential direction, especially those with new strategies or new targets.
- The preclinical development using new strategies and targets is also missing.
- TCR-T therapies not mentioned.
minor point:
- first line: Reiew --> Review
- Page 4: due --> do
Author Response

(The authors gave the same response as above.)

Reviewer 3 Report
Comments and Suggestions for Authors
This paper provides a comprehensive review of novel T-cell-directed therapies for patients with relapsed/refractory aggressive and indolent Non-Hodgkin Lymphomas (NHL). The authors offer a detailed discussion of various clinical trials involving CAR T-cells and bispecific antibodies.
This is a timely and relevant review that highlights both the benefits and challenges of these emerging therapies. Overall, the paper is well-written, but some formatting issues should be addressed.
Currently, the paper includes only two tables summarizing CAR T-cell therapy outcomes for DLBCL, with no corresponding data for other lymphoma types. It would be beneficial to include additional tables summarizing results for other NHLs. Furthermore, tables should be integrated within the main text rather than placed at the end of the manuscript.
Additionally, a comparative table outlining the key differences between CAR T-cell therapies and bispecific antibodies would enhance the paper's clarity and usefulness.
To improve readability, the text should be better structured by dividing long paragraphs into clearly defined sections.
Author Response

(The authors gave the same response as above.)
